# First Steps to Domesticate Hairy Stork's-Bill (*Erodium crassifolium*) as a Commercial Pharmaceutical Crop for Arid Regions

**Ofer Guy** [1,*], **Shabtai Cohen** [1], **Hinanit Koltai** [2] , **Moran Mazuz** [2], **Moran Segoli** [1] **and Amnon Bustan** [1]

1 Ramat Negev Desert Agro-Research Center (RNDARC), Ramat Negev Works Ltd., D.N. Halutza 8551500, Israel; sab@inter.net.il (S.C.); moran.segoli@gmail.com (M.S.); amnonbustan@gmail.com (A.B.)
2 Department of Ornamental Horticulture and Biotechnology, Institute of Plant Sciences, Agricultural Research Organization, Rishon LeZion 7528809, Israel; hkoltai@volcani.agri.gov.il (H.K.); moranjacobi@gmail.com (M.M.)
* Correspondence: oferguy243@gmail.com; Tel.: +972-50-3699001

**Abstract:** Hairy stork's-bill (*Erodium crassifolium*) (HSB) is one of the few Geraniaceae species that produce tubers. While HSB tubers were occasionally used as a food source by desert nomads, they have not yet been taken up in the modern kitchen. Recently, HSB tubers were recognized as harboring potential to become an industrial pharmaceutical crop. The objective of this study was to determine a set of agricultural practices that would maximize the yield of the bioactive compounds of the present HSB genetic material. A generous and consistent irrigation regime of about 700 mm season$^{-1}$ increased tuber yield and size. The optimal plant density on sandy soils was 16–20 plants m$^{-2}$. Mineral nutrition, applied through fertigation, increased tuber yield from 0.6 to almost 1 kg m$^{-2}$. Source-sink manipulations (consistent pruning of the reproductive organs) almost doubled the tuber yield. During 10 years of research, the adopted agricultural practices increased yield by an order of magnitude (from about 0.2–2.0 kg m$^{-2}$) without any dilution of the bioactive compounds. Nevertheless, further research and development are required to achieve HSB potential as an industrial field crop, including selection and breeding of outstanding infertile clones, optimization of fertigation, and development of various concrete pharmaceutical products.

**Keywords:** arid land; *Erodium crassifolium*; medicinal crops; mineral nutrition; source-sink relations; underutilized species; plant domestication

## 1. Introduction

Hairy stork's bill (HSB) (*Erodium crassifolium* L'Hér) is a Saharo-Arabian perennial hemicryptophyte (i.e., buds are at or near the soil surface) common in shrub-steppes of arid southeast Mediterranean regions. The species is distributed from northwest at Crete [1], through few Aegean Sea islands [2], the Libyan [3] and Egyptian coasts [4,5], north Sinai Peninsula [4,6], Cyprus [7], the Negev Desert of Israel [8], until Edom mountains of Jordan and Saudi-Arabia deserts, on southeast [9]. In Israel, HSB is found in the Negev and Judean deserts, where the annual rainfall is in the range of 30–250 mm. The species is most abundant in the stony and arid loess soils and on the slopes of limestone hills [8]. There are several accessions of HSB ecotypes in the Israeli plant gene bank at the Agricultural Research Organization (ARO), however, differences between ecotypes have not been characterized. The life cycle of HSB is described in Table 1. Germination begins after the first effective rain event, and an early vegetative phase occurs, forming typical rosette leaves. During spring, further vegetative and reproductive growth waves occur, the intensity and size of which depend on water availability [10]. Hairy stork's bill is among the few plants of the Geraniaceae family to produce tubers. The first tubers occur soon after germination and thereafter, during the vegetative phase. The tubers are located on roots at a depth of 5–20 cm and are typically small and spherical (1–2 cm in diameter). Bedouin ethnobotanical

wisdom holds that the tubers are edible and as such local Bedouin tribes are their primary users [8]. HSB tubers have a light sweet taste, and their best quality is in late winter or early spring, when they are whitish in color [8].

**Table 1.** Life cycle and phenological development of HSB as observed in our irrigated fields.

| Month | December | | January | | | | February | | | | March | | | | April | | | | May | | | June | |
|---|---|---|---|---|---|---|---|---|---|---|---|---|---|---|---|---|---|---|---|---|---|---|---|
| **Week of Year** | 51 | 52 1 | 2 | 3 | 4 | 5 | 6 7 | 8 | 9 | 10 | 11 12 | 13 | 14 | 15 | 16 17 | 18 | 19 | 20 21 | 22 | 23 24 | | | |
| Developmental stages | Imbibition | | | | | | | | | | | | | | | | | | | | | | |
| | | | Germination | | | | | | | | | | | | | | | | | | | | |
| | | | | | | Vegetative | | | | | | | | | | | | | | | | | |
| | | | | | | | | Tubers production | | | | | | | | | | | | | | | |
| | | | | | | | | | | | Bloom | | | | | | | | | | | | |
| | | | | | | | | | | | | | | Fruiting + seed dispersal | | | | | | | | | |
| Irrigation | 6　mm day$^{-1}$ | | | | | 4 mm day$^{-1}$ | | | | | | | | | | | | | | | | | |
| (mm month$^{-1}$) | 84 | | 152 | | | 120 | | | | 124 | | | | 120 | | | | 80 | | | | | |

While their culinary appeal is somewhat dull, recent studies in our labs have demonstrated that HSB tubers harbor significant pharmaceutical potential [10,11]. Electrochemical analysis of the water-soluble extract from HSB tubers revealed significant reducing power and identified at least six groups of antioxidants (data not shown). Fractionation of the tubers' ethanolic extract (EE) and subsequent *in-vitro* evaluations of possible antiinflammatory capacities revealed significant activity in fractions F3 and F4, as well as in the original EE and the pooled fractions, PF [10,11]. Biochemical analyses showed that F4 comprised mainly epigallocatechin, trans- and cis-catechin, and gallic acid, all of which are known for their robust bioactive capacities, including anti-inflammatory activity [12–18]. Interestingly, the bioactivity of F4 was significantly greater than that of each compound alone at its corresponding concentration [11] indicating synergic relationships in the natural extract. Furthermore, recently we have shown that HSB tubers' extracts displayed significantly stronger anti-inflammatory capacity compared to extracts of green tea (*Camellia sinensis*) and turmeric (*Curcuma longa*) [11], well-known sources of anti-oxidative and bioactive compounds [12,19].

An essential demand for the domestication of a given plant species must begin with a recognition of significant potential benefits that it may provide. Later steps include, and not necessarily in the following order, selection for outstanding ecotypes, breeding, and designing a set of environmental growing conditions that should maximize the benefits. While in ancient times this procedure could have lasted centuries [20–23], nowadays it can be significantly shortened. Indeed, the phase of genetic selection and breeding might be long, as it depends on numerous yearly cycles of experimental efforts. Nevertheless, evaluating the potential of new species and suiting a set of optimal environmental conditions might be much shorter, due to enhanced scientific capacities (e.g., comprehensive chemical and pharmaceutical analyses) and technological progress (e.g., controlled drip fertigation).

HSB is still a wild plant species; so far, no other attempts have been made to domesticate it. In an attempt to realize HSB pharmaceutical potential [10,11], the objective of this study was to determine a set of environmental conditions, namely, agricultural practices, that would maximize the yield of the bioactive compounds of the given genetic material. Assuming that HSB is an opportunistic plant species in arid environments (30–250 mm season$^{-1}$), crop performance was examined under substantially generous terms of water (600–700 mm season$^{-1}$) and mineral fertilizer supplies. Additionally, plant density was optimized, problems of pests and soil degradation were identified and managed. Hypothesizing that competition between developing tubers and reproductive organs might occur, the response of the tuber yield and the concentrations of bioactive compounds to source-sink manipulations (pruning of reproductive organs) was exam-

ined. Unequivocally, however, the agricultural practices should be further improved and adjusted according to prospective changes in HSB genetic material.

## 2. Materials and Methods

### 2.1. Ecosystem and Plant Material

The research took place at Ramat Negev Desert Agro-Research Center (RNDARC), Israel (30°58′ N 34°42′ E), 305 m above sea level. Seeds of Hairy stork's-bill plants were sown in sandy Soil beds and irrigated in an open field-block designed experiments. The life cycle of HSB in its natural habitat takes place in the rainy season, from November to May. Mean daily temperature declines from about 18 °C in November to about 10 °C in January, and then steadily rises to 23 °C in May. Minimum temperatures vary from 2.4–12.8 °C from January to May, respectively, while maximum temperatures range from 17.4–32.9 °C for the same period, respectively. However, considerable variations in the monthly mean extreme temperatures occur between years, generating large differences in the seasonal temperature regime between years (Table 2). These differences are even greater with respect to the precipitation regime (Table 2). While the mean annual precipitation is about 90 mm, substantial year-to-year as well as local variations prevail regarding rainfall amounts, frequency, and intensity, all of which have significant consequences on HSB life span in a given year.

All experiments were conducted using seeds of the ecotype Ramat Negev (RN). Four additional ecotypes were examined in comparison.

### 2.2. Field Experiments

The data presented in this study were obtained in two sets of field experiments, each conducted for two years at RNDARC from November to June in 2009–2011 and 2016–2018. In the first set, crop performance was studied in response to plant density, while the second set examined various agricultural practices aimed at increasing yield.

In the Negev desert, HSB usually inhabits loess soils with various sand proportions. In this study, sandy soils were used since tuber harvest is easier in sandy soil compared to loess soil [24]. Compost was applied at 42 ton ha$^{-1}$ before planting, as a soil amendment [25].

HSB seeds were sown in mid-December on local agricultural sandy soil beds. Distances between beds were 1.2 m. Two seeds were sown in a shallow seed nest in four separated rows in the middle of the bed. Sowing on both sides along the drip line every 20 cm resulted in 20 seed nests per meter bed (17 seed m$^{-2}$) and ensured seeds proximity to emitters. According to the prevalent case, all plants arising from a seed-nest were considered a single plant. A 17 mm drip line with integrated emitters every 20 cm was used with flow rate of 1.6 liter hour$^{-1}$. During the 2009–2011 seasons, experiments were conducted in a net house (50 mesh), while the rest of the experiments were conducted in open field conditions. All field experiments were conducted using a randomized complete block design (RCBD) with four replicates (plot size was 2.4 m$^2$).

### 2.3. Fertigation

Irrigation with water without fertilizer (0.7 dS m$^{-1}$), was executed at a rate of 6 mm day$^{-1}$ until 50% germination, about 3–4 weeks after sowing. Then, irrigation was reduced to 4 mm day$^{-1}$ and fertilizer application commenced. Usually, fertigation was practiced using a liquid N-P-K composite fertilizer (Shefer 4:2:6, Israel Chemicals Ltd., Tel Aviv-Yafo, Israel) at concentration of 1.5 liter m$^{-3}$, and at nitrate: ammonium ratio of 2:1, respectively. Irrigation was ceased upon significant decline of plant vigor towards the end of May, with a total annual mean of about 700 mm (Table 1).

### 2.4. Plant Density Trials (2009–2011)

Two experiments were carried out in the seasons 2009–2010 and 2010–2011 in order to examine the effects of plant density on crop performance. In the first season, seed-

nests were located along both sides of the dripline in spaces generating four distinct plant densities: 4, 8, 16, and 20 plants m$^{-2}$. In the consecutive season, in order to verify previous results and extend the examined range, three plant densities were tested: 16, 20, and 24 plants m$^{-2}$.

**Table 2.** Monthly mean minimum and maximum temperature, and monthly precipitation recorded at RNDARC during the recent 12 HSB growing seasons from October to May.

| | Season | Month | | | | | | | | |
|---|---|---|---|---|---|---|---|---|---|---|
| | | October | November | December | January | February | March | April | May | |
| Minimum temperature (°C) | 2009/10 | 15.7 | 9.1 | 7.3 | 7.0 | 8.7 | 10.4 | 10.5 | 12.7 | |
| | 2010/11 | 15.6 | 11.2 | 6.1 | 4.3 | 7.0 | 5.6 | 9.8 | 13.2 | |
| | 2011/12 | 12.0 | 5.0 | 2.7 | 1.8 | 4.7 | 6.1 | 9.6 | 13.4 | |
| | 2012/13 | 14.2 | 11.4 | 5.6 | 5.4 | 7.0 | 9.1 | 10.5 | 14.6 | |
| | 2013/14 | 10.2 | 10.8 | 5.2 | 4.6 | 4.2 | 9.2 | 10.8 | 14.7 | |
| | 2014/15 | 12.4 | 10.3 | 6.6 | 4.7 | 5.2 | 8.2 | 9.0 | 13.7 | |
| | 2015/16 | 16.2 | 12.1 | 3.7 | 3.8 | 5.9 | 8.7 | 11.5 | 14.4 | |
| | 2016/17 | 13.6 | 10.3 | 3.9 | 2.4 | 3.2 | 7.3 | 9.5 | 12.8 | |
| | 2017/18 | 12.4 | 7.7 | 6.8 | 5.7 | 8.2 | 8.4 | 10.7 | 17.0 | |
| | 2018/19 | 16.0 | 10.3 | 6.5 | 3.0 | 5.4 | 6.2 | 8.9 | 13.4 | |
| | 2019/20 | 15.0 | 10.4 | 5.1 | 4.4 | 6.2 | 7.5 | 10.4 | 13.7 | |
| | 2020/21 | 16.3 | 10.1 | 7.0 | 5.7 | 6.8 | 7.0 | 9.7 | 14.4 | |
| | Mean | 14.1 | 9.9 | 5.5 | 4.4 | 6.0 | 7.8 | 12.7 | 14.0 | |
| | SD | 2.00 | 1.90 | 1.47 | 1.49 | 1.61 | 1.45 | 0.78 | 1.16 | |
| Maximum temperature (°C) | 2009/10 | 32.1 | 23.7 | 20.8 | 21.5 | 22.4 | 25.9 | 28.4 | 32.1 | |
| | 2010/11 | 32.8 | 29.6 | 22.5 | 19.2 | 19.9 | 23.0 | 26.9 | 30.8 | |
| | 2011/12 | 29.1 | 21.8 | 19.8 | 16.5 | 18.5 | 21.0 | 29.2 | 32.0 | |
| | 2012/13 | 31.1 | 25.5 | 20.4 | 18.1 | 20.6 | 26.3 | 27.4 | 33.2 | |
| | 2013/14 | 28.8 | 26.7 | 18.7 | 19.4 | 21.1 | 24.1 | 29.6 | 31.5 | |
| | 2014/15 | 29.3 | 23.6 | 21.9 | 17.2 | 19.2 | 23.7 | 27.0 | 32.8 | |
| | 2015/16 | 30.9 | 24.6 | 18.7 | 17.0 | 22.7 | 24.8 | 31.3 | 32.4 | |
| | 2016/17 | 31.5 | 25.3 | 17.8 | 17.4 | 18.6 | 24.1 | 28.6 | 32.9 | |
| | 2017/18 | 29.4 | 24.5 | 22.2 | 17.5 | 23.3 | 28.0 | 29.0 | 34.4 | |
| | 2018/19 | 30.5 | 24.4 | 19.4 | 18.6 | 19.4 | 21.0 | 26.5 | 34.7 | |
| | 2019/20 | 30.9 | 26.9 | 20.3 | 16.3 | 18.7 | 22.8 | 27.5 | 33.3 | |
| | 2020/21 | 32.4 | 24.5 | 22.2 | 19.7 | 21.7 | 23.7 | 29.6 | 34.5 | |
| | Mean | 30.7 | 25.1 | 20.4 | 18.2 | 20.5 | 24.0 | 28.4 | 32.9 | |
| | SD | 1.35 | 1.97 | 1.57 | 1.53 | 1.71 | 2.05 | 1.41 | 1.22 | |
| | | | | | | | | | | Sum |
| Precipitation (mm month$^{-1}$) | 2009/10 | 32.7 | 4.5 | 16.7 | 59.7 | 30.3 | 23.0 | 0.2 | 1.2 | 168.3 |
| | 2010/11 | 0.0 | 0.9 | 2.8 | 5.5 | 12.7 | 2.3 | 3.1 | 1.3 | 28.6 |
| | 2011/12 | 0.4 | 5.6 | 0.0 | 17.2 | 8.4 | 27.6 | 0.0 | 0.1 | 59.3 |
| | 2012/13 | 0.0 | 17.9 | 4.2 | 80.1 | 15.2 | 0.1 | 0.0 | 10.3 | 127.8 |
| | 2013/14 | 10.6 | 2.3 | 25.8 | 4.6 | 4.1 | 47.1 | 0.1 | 12.1 | 106.7 |
| | 2014/15 | 6.1 | 12.6 | 15.4 | 36.1 | 49.3 | 10.6 | 10.8 | 0.0 | 140.9 |
| | 2015/16 | 19.2 | 21.8 | 12.5 | 16.4 | 21.0 | 4.1 | 23.7 | 0.0 | 118.7 |
| | 2016/17 | 6.2 | 0.4 | 20.6 | 8.1 | 17.3 | 1.2 | 2.3 | 0.0 | 56.1 |
| | 2017/18 | 0.1 | 4.8 | 3.7 | 54.6 | 26.3 | 0.5 | 1.7 | 0.3 | 92.0 |
| | 2018/19 | 0.5 | 42.5 | 4.3 | 0.5 | 31.5 | 25.1 | 2.5 | 0.0 | 106.9 |
| | 2019/20 | 13.1 | 0.4 | 3.4 | 37.1 | 40.3 | 39.9 | 1.1 | 6.9 | 142.2 |
| | 2020/21 | 0.1 | 9.1 | 6.5 | 5.5 | 16.6 | 0.8 | 0.0 | 0.0 | 38.6 |
| | Mean | 7.4 | 10.2 | 9.7 | 27.1 | 22.8 | 15.2 | 3.8 | 2.7 | 98.8 |
| | SD | 10.2 | 12.3 | 8.3 | 26.2 | 13.3 | 16.8 | 6.9 | 4.4 | 44.6 |

*2.5. Ecotype Examination (2016–2018)*

Two experiments were conducted in the seasons of 2016–2017 and 2017–2018, in order to compare between the five ecotypes. In addition to RN ecotype, we evaluated Revivim ecotype, the seeds of which were collected annually at the vicinity of Kibbutz Revivim, and three HSB accessions (ecotypes 25525, 25493, and 26164) received from the Israeli Gene Bank, ARO.

### 2.6. Fertilizer Trials (2016–2018)

Until 2016, HSB requirements for mineral nutrition were supplied through fertigation, as described above, based on the local recommendation for high-yielding potato crops [26]. From 2016 to 2018, two experiments were carried out to examine possible effects of a different ammonium: nitrate ratio, 1:9, in the nitrogen fraction of the fertilizer, as an alternative to the former ratio of 1:2. These experiments also included a non-fertilized control.

### 2.7. Source-Sink Manipulation (2016–2018)

Altering source-sink relations were suspected to affect HSB tuber yields. Two consecutive experiments were carried out in the seasons 2016–2017 and 2017–2018 aiming at examining this assumption using a recurrent hand-pruning of the reproductive organs. These experiments included four treatments that differed in the launching time of pruning, as follows: control-no pruning; bloom-weekly removal of all inflorescences from the first week of April, at the earliest emergence of a considerable bloom; capsule–biweekly removal of all reproductive organs, beginning on mid-April, at the initiation of fruit set; seeds–biweekly removal of all reproductive organs, beginning on early May, at the initiation of seed dispersal.

### 2.8. Soil Recovery (2020–2021)

In 2020, the experiments focused on soil degradation. Consecutive HSB cultivation on the same plots resulted in yield reduction. In order to overcome the problem, an experiment was conducted in a field that had experienced HSB cropping in the 2019–2020 season. The experiment included three soil treatments: a. soil disinfection, combining solar heating using transparent polyethylene sheets, and separate applications of Metam-sodium and of 1,3 Dichloropropene 93% [27]; b. (control)-no treatment; and c. green amendment of mustard (type Terminator, Genesis organic seeds Ltd.), grown and embedded in the soil. All treatments were composed of two beds, 15–20 m long. Plants were fertigated and grown in the same conditions described above. Compost was added to all treatments at 42 ton ha$^{-1}$ prior to sowing.

### 2.9. Measurements

Crop phenology was followed and documented in all experiments, as shown in Table 1. Tuber harvest took place in June, along with seeds ripening and dispersal. At harvest, tuber yield parameters, such as tubers number (tuber plant$^{-1}$; tubers m$^{-2}$), mean tuber weight, and tuber yield (g plant$^{-1}$; t ha$^{-1}$) were recorded. Tubers were cleaned and sorted into four categories: S1—very young, small, bright-skin tubers; S2—young, fully-grown, bright-skin tubers; S3—maturing, brownie-skin tubers; and, S4—senescing, dark-skin tubers (Figure 1). Preliminary studies showed that only S2 and S3 tubers were relevant for the extraction of bio-active compounds and hence, these were freezed and stored at −20 °C, until further analyzation.

### 2.10. Ethanolic Extraction (EE) of HSB Tubers

HSB tubers were removed from cold storage (−20 °C) and frozen in liquid nitrogen. The frozen tubers were crushed using an electrical blender and weighed. For each 1 g of fresh material, 4 mL of 70% ethanol were added immediately to the crushed tubers and incubated overnight at 28 °C with shaking at 180 rpm, after which the samples were centrifuged for 5 min at 5000 rpm. The supernatant was transferred to new tubes. The solvent was evaporated *in vacuo* overnight. The remaining water content was lyophilized to powder and stored at −20 °C. From each gram of tubers, approx. 60 mg of lyophilized extract was obtained.

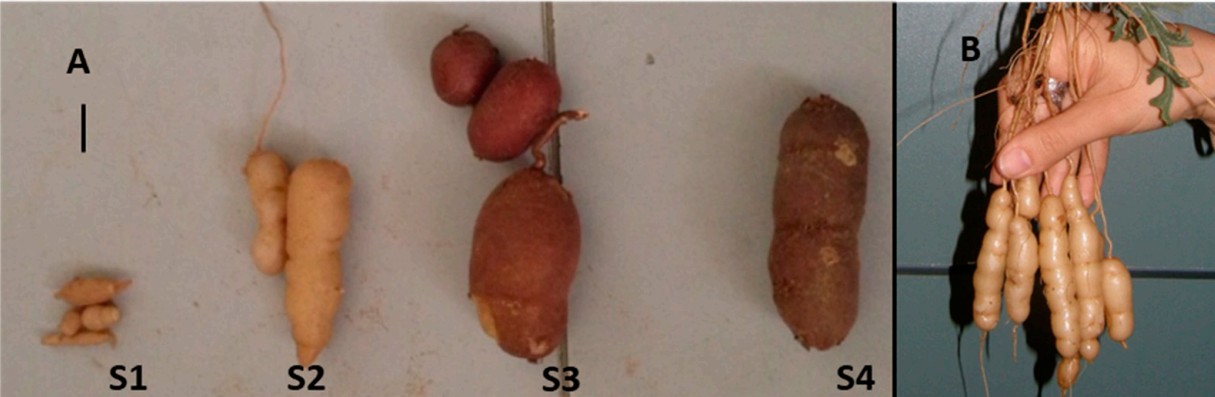

**Figure 1.** Age-related changes in size and morphology of harvested HSB tubers (**A**): S1—very young, small tubers; S2—young, fully grown tubers; S3—mature tubers with suberized skin; S4—an old, deep brown tuber. Bar indicates 1 cm. A typical beady elongated HSB tubers harvested from irrigated plots (**B**).

Just before further analysis, the lyophilized material was weighed and dissolved in 100μL of 70% ethanol and then 900 μL of double distilled water (DDW), to obtain a 60 mg mL$^{-1}$ sample, which was filtered through a 0.45 μm membrane. The filtrate was diluted or not for further testing as described below.

*2.11. Human Cell Culture and IL-8 Determination*

HCT116 (ATCC CCL-247) colon cell line was grown at 37 °C in a humidified 5% $CO_2$-95% air atmosphere. Cells were maintained in McCoy's 5a Modified Medium. Cells were seeded, in triplicate, into a 24-well plate at a concentration of 50,000 cells per well in 500μL of growing media, and then incubated for 24 h at 37 °C in a humidified 5% $CO_2$-95% air atmosphere. When cell excitation was performed with TNFα, cultures in each well were treated with 50 ng/mL recombinant human TNFα (PeproTech, Rocky Hill, NJ, USA) with or without HSB tuber extract. The supernatant was taken and the levels of IL-8 were measured per 1 g of dry weight of HSB tuber crude extract at 16 h post-treatment using the commercial Human CXCL8/IL-8 DuoSet ELISA kit (R&D Systems, Minneapolis, MN, USA). In each *in-vitro* experiment, a Tukey-Kramer HSD test ($p \leq 0.05$; $n = 3$) was executed. The results were normalized to the TNFα control and expressed in relative units (%).

**3. Results and Discussion**

*3.1. Water Availability Determines Crop Productivity*

In the Negev desert, the rainy season (October-April) is characterized by low annual precipitation, ranging from 30–250 mm (Table 2), which is distributed over few sporadic rain events. In the wild, HSB thrives better at local micro catchments of water runoff during rain events, where the water availability is by far greater than the measured periodic precipitation [24]. As a major limiting factor, water availability in its natural habitat determines HSB's growth and tuber production capacity. Moreover, as a desert plant, its germination is sporadic and asynchronous, properties that might hinder its utilization as a potential crop [22,23]. To overcome such obstacles, two seeds per seed-nest were sown, and a continuous routine of daily irrigation was employed for 3–4 weeks (Table 1). This procedure resulted in high soil water content and finally, provided high germination rate of about 95%.

Following the natural timing of HSB germination, seeds were regularly sown in late December. Notably, December and January are usually the coldest months of the year (Table 2) and therefore, the prevailing low temperatures might be the reason for the long duration of imbibition and germination under field conditions (Table 1). Using nursery seedlings may shorten the period of crop establishment as well as save about 25% of the annual amount of irrigated water.

In the wild, the life span of seedlings and consequently, their vegetative and reproductive biomass, directly depend on an adequate and consistent water availability. Subsequently, irrigation management is a predominant tool in maximizing tuber yield and quality. HSB tubers initiate and develop very early, shortly after the vegetative growth commences, already in February, and they continue to emerge and grow as long as adequate water supply endures, until June (Table 1). The number, size, and shape of the tubers are a function of the stability of the water supply [10,11]. Interruptions of the water supply induce transient cessation of tuber growth, which might lead to abnormal tuber shapes, tuber senescence (Figure 1B), and at the worse cases, to death.

### 3.2. Crop Density

In the wild, HSB plant size, and the number and size of the tubers it produces, are substantially diverse, determined by the local availability of water, nutrients, and space. Competition with neighboring plants on the aboveground space restricts light penetration into the canopy, thus reducing plant's carbon assimilation and primary production. Neighboring plants also may compete at the belowground space, particularly in tuber-producing species. Optimizing crop density is among the first steps required to maximize yield and quality.

The number of tubers produced by an individual HSB plant significantly declined by 50%, when crop density was increased from 4 to 16 plants $m^{-2}$ (Figure 2A). In contrast, the number of tubers per area was doubled (Figure 2B). Further increase in crop density had much smaller effects, as indicated in 2010 at crop density of 20 plants $m^{-2}$, and in 2011, at a crop density range of 16–24 plants $m^{-2}$. Interestingly, the number of tubers was considerably smaller in 2011 (Figure 2A,B). The response of tuber yield of individual plants to increasing crop density strictly followed that of the number of tubers (Figure 2C), indicating almost no compensation occurring between the number of tubers and the mean tuber weight. Bringing together the results of both experiments revealed a crop density optimum range of 16–20 plants $m^{-2}$, at which maximum tuber yield is obtained (Figure 2D). Consequently, a crop density of 17 plants $m^{-2}$ is practiced in experiments as well as commercial pilot fields.

### 3.3. Mineral Nutrition

Desert soils, and particularly sandy soils, are usually poor in mineral nutrients and therefore affected by addition of mineral nutrition (Figure 3; Wilks' lambda = 0.26, $F_{(6,23)} = 3.53$, $p = 0.01$). Addition of NPK fertigation throughout the season (300, 190, and 450 kg $ha^{-1}$ of N, $P_2O_5$, and $K_2O$, respectively) gave rise to about 60% increase in tuber yield, compared to the control that did not receive fertilization (Figure 3A). While the ratio of ammonium to nitrate had no significant influence on the total tuber yield (Figure 3A), a higher proportion of nitrate tended to increase the S2 + S3 tuber fraction in the expense of the older S4 fraction, as compared to the higher proportion of ammonium fertilizer (Figure 3B).

In the present study, HSB tuber yields substantially increased in response to N-P-K doses of 300–400 kg N, 190–280 kg $P_2O_5$, and 450–530 kg $K_2O$ per ha. These high levels of fertilizer input were adopted from the protocol for highly productive potato (*Solanum tuberosum*) grown on poor sandy soils, where consistent fertigation is a must [26,28].

Obviously, the current nutrient inputs cannot be justified neither by the actual tuber yield level, nor by the caloric value of the tubers [10,11]. In spite of the substantial nutrient inputs, quite limited attainment was produced, with a ceiling tuber yield of less than 10 t $ha^{-1}$. Both water and fertilizer application practices require a considerable reassessment in order to substantively increase the agronomic use efficiencies of these basic inputs.

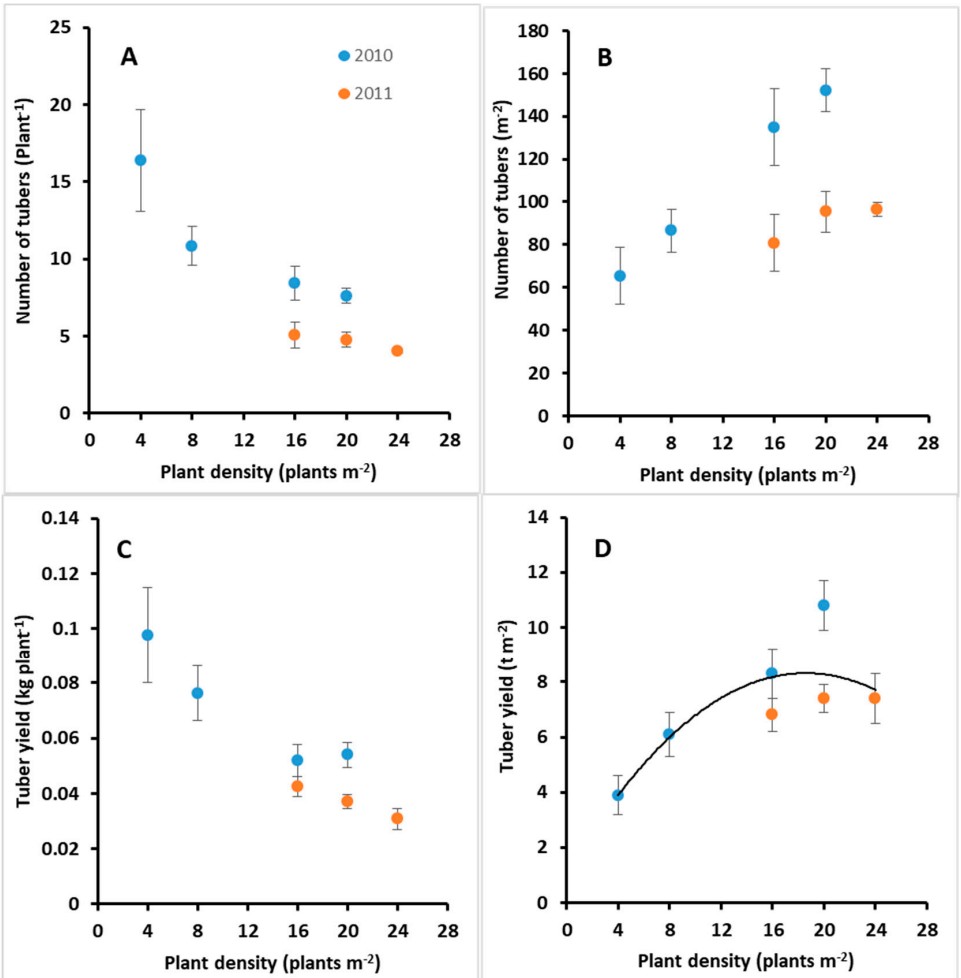

**Figure 2.** Effects of plant density on the number of HSB tubers per plant (**A**) and per area (**B**), and on the tuber yield per plant (**C**) and per area (**D**). The experiments were conducted in 2010 and 2011. Bars indicate SE.

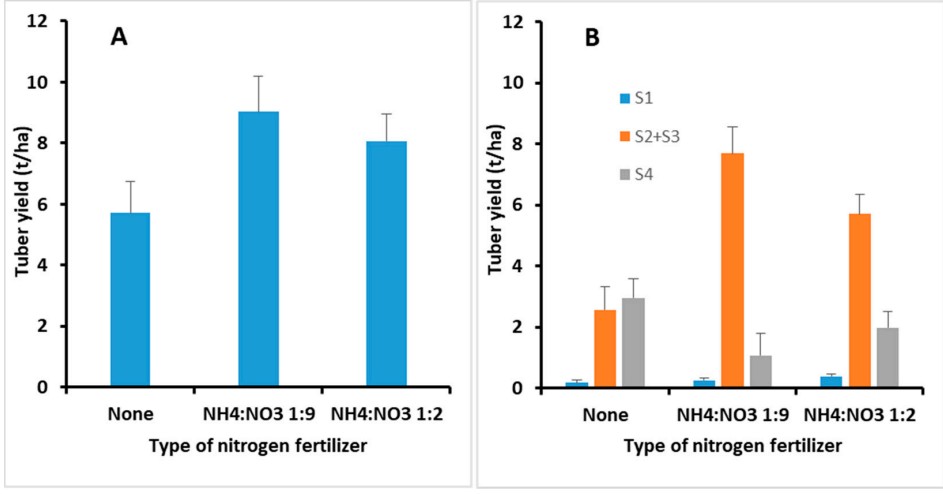

**Figure 3.** Effects of composite N-P-K fertilizer application (seasonal doses of 300, 190, and 450 kg ha$^{-1}$ of N, $P_2O_5$, and $K_2O$, respectively), comparing two different ammonium:nitrate ratios on total tuber yield (**A**), and on tuber distribution according to size, quality, and age (**B**). Tubers were sorted according to Figure 1. Bars indicate SE.

### 3.4. Source-Sink Manipulations

Tubers are continuously produced by HSB plants almost throughout the vegetative phase, as long as adequate water availability sustains. The reproductive phase begins in late March with apical inflorescences, however, new branches soon emerge from auxiliary buds and later bear new flowers, thus forming a continuous dense bloom. In parallel, flowers set fruit (capsule) and seeds, forming the typical stork-bill. It appears that, in addition to the prosperous vegetative phase, the supportive agricultural environment substantially promotes an immense reproductive effort. Under simultaneous production of reproductive organs and tubers, a competition on dry matter allocation may occur [29].

Source-sink manipulation increased tuber yield (one-way ANOVA $F_{(3,16)} = 10.13$, $p < 0.01$). When the inflorescences were recurrently pruned during the bloom stage, tuber yield surged to almost 2 kg m$^{-2}$ (20 t ha$^{-1}$), twice as high as control. Later interventions in the reproductive phase resulted in positive, though smaller effects (Figure 4A). Pruning at bloom also doubled the valuable tuber fraction (S2 + S3) from 0.7 to 1.54 kg m$^{-2}$ (Figure 4B). In most of the commercial potato cultivars, bloom is infertile and hence, fruit are not produced. Thus, during the breeding of potatoes, a competition between tubers and fruits was avoided through the preference of infertile cultivars [29]. Considering the significant effect of the reproductive pruning in HSB, selection and breeding efforts should be focused on flower infertility.

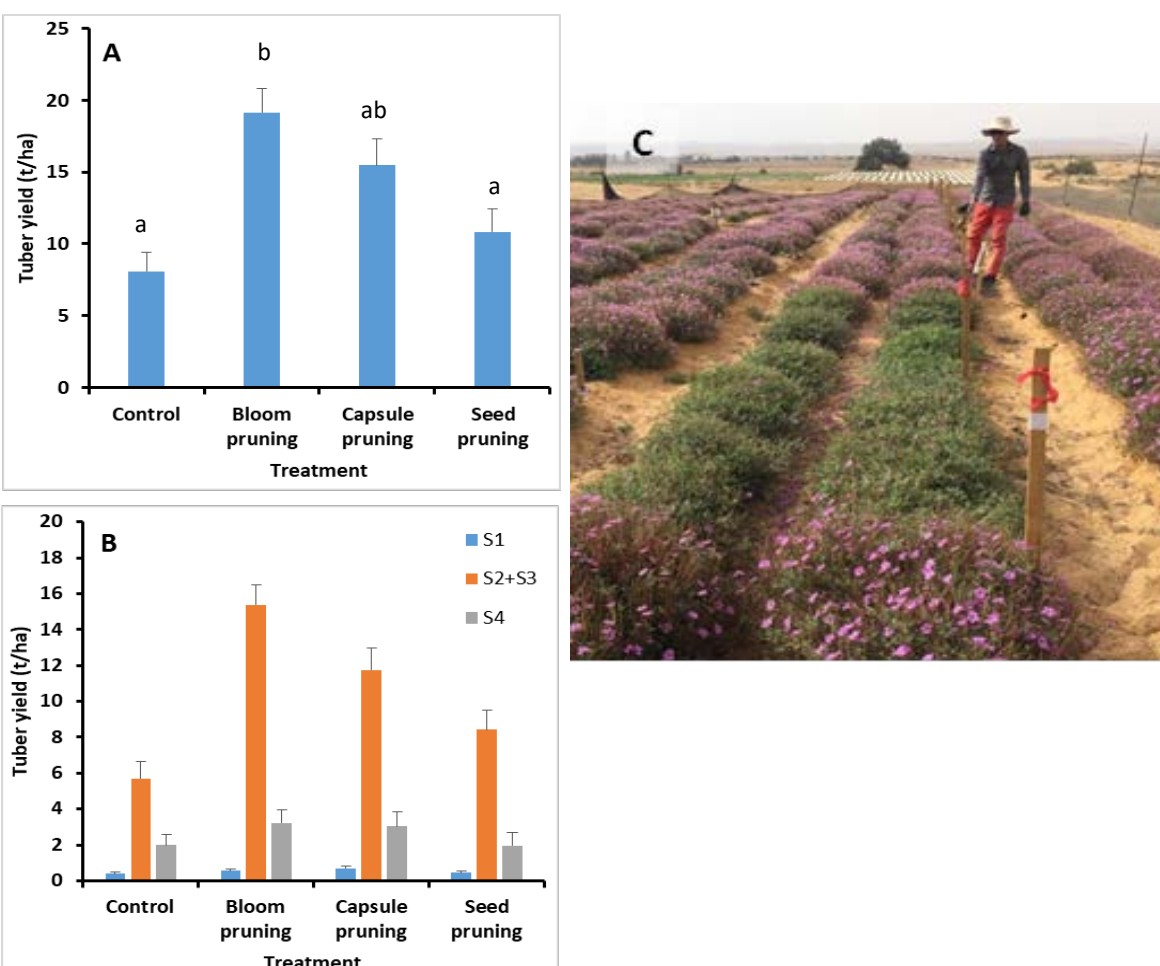

**Figure 4.** Effects of recurrent pruning of reproductive organs during three consecutive developmental stages: bloom, fruit (capsule) set, and seed formation on the total HSB tuber yield. Different superscript letters indicate significant differences (**A**), and on the distribution of the tubers between fractions according to size, quality, and age (**B**). A picture demonstrating pruning treatments (**C**). Control is without pruning at all. Bars indicate SE.

### 3.5. Anti-Inflammatory Activity

HSB tubers harbor various beneficial bioactive ingredients [10,11]. IL-8 is a well-known inflammation marker, expressed during inflammation in colon epithelial cells [30]. Unsurprisingly in this respect, significant differences in reduction of IL-8 levels were found between the few HSB ecotypes examined in the present study. While ecotypes Ramat Negev, Revivim, and 25525 exhibited anti-inflammatory activity (in vitro), ecotype 25493 had a much smaller effect, and ecotype 26164 was hardly effective (Figure 5A). These results imply that the therapeutic potential of this species should be explored among ecotypes. Furthermore, ecotype selection and genetic breeding must be employed to place HSB among considered industrial medicinal crops.

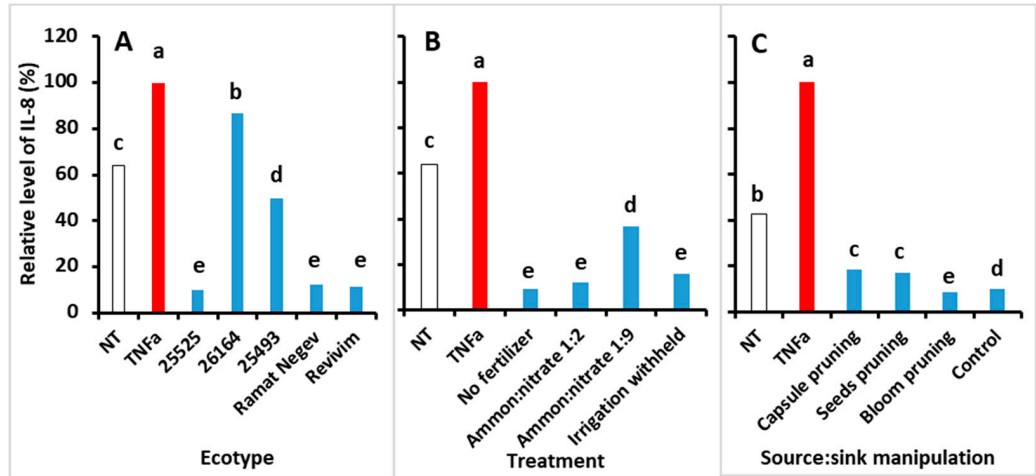

**Figure 5.** The relative level of IL-8 in HCT-116 cells treated with HSB tuber extracts of in different ecotypes (**A**), different mineral nutrition (**B**) and source-sink manipulations (**C**). Cells were treated with 50 ng/mL TNFα with or without HSB tuber extracts for 16 h. NT is the solvent (vehicle) control (7% *v/v* ethanol); TNFα is TNFα + solvent control treatment. Bars labeled with different letters are significantly different from all combinations of pairs according to the Tukey-Kramer HSD test ($p \leq 0.05$; $n = 3$).

Agricultural practices may affect the content of bioactive compounds in HSB tubers. When the ammonium:nitrate ratio was reduced from 1:2 to 1:9, the *in-vitro* anti-inflammatory activity, as determined by inhibition of IL-8 production in excited HCT-116 cells, was significantly reduced (Figure 5B). Inflammatory processes in the colon epithelial are closely related to increased expression of interleukin (IL)-8, in addition to tumor necrosis factor alpha (TNF-α) and phosphorylated NF-κB (nuclear factor κB) p65. High levels of these proteins were identified in inflamed epithelia of ulcerative colitis patients [30]. In contrast, an earlier cessation of the irrigation (two weeks before usually done) promoted an earlier tuber senescence, as indicated by a darker tuber color, but did not reduce the tubers anti-inflammatory activity (Figure 5B). Source-sink manipulations resulted in a significant effect on the anti-inflammatory activity of HSB tubers (Figure 5C); while an early and recurrent removal of all the reproductive organs (bloom pruning) enhanced tuber anti-inflammatory activity. Similar but later and less extensive interventions (capsule or seed pruning) had an opposite effect, with lower anti-inflammatory capacity, compared to the unpruned control. While the reason for these differences remains obscure, these results indicate that the increment in HSB tuber yield should not be the sole parameter to evaluate agricultural practices; these might have contradictory, additive or synergistic influences on the bioactive compounds that should be carefully considered and deserve much further research.

### 3.6. Challenges and Solutions

Three major challenges emerged during this HSB research, most of which were resolved: tuber harvesting; pests; and soil degradation.

HSB are located at a depth of 5–25 cm of the soil profile. Mechanical harvesting methods were successfully adopted from potatoes, with no considerable hazards to tuber quality.

While no serious foliar pests or diseases could be observed during more than 10 years of experiments, the tubers can be severely damaged by the rose beetle *Maladera insanabilis*, which is known to attack a wide range of plant families [31]. The larvae of this insect live in the soil and feed by chewing the young roots of crops such as sweet potato, peanuts, and strawberry plants [32], causing deterioration of plants and leading to an inevitable death, depending on the larva density and plant age [33]. In 2018, a *M. insanabilis* attack resulted in significant losses of HSB tubers, mainly at the outskirts of the experimental plots. In the subsequent years, 200 g/L Chlorantraniliprole (Coragen) was applied once a season via the irrigation system about 50 days before harvest [34]. The pest did not reoccur, and no detected chemical residuals were found in the tubers when harvested.

When HSB was grown on the same plot over two consecutive years, tuber yields significantly declined. Seed germination was substantially delayed, and its rates declined, leading to an extremely patchy pattern of soil coverage and subsequently, to reduced tuber production. While the precise reason for this phenomenon is not known, regular practices of soil disinfection, combining solar heating with transparent polyethylene sheets, Metam-sodium, and separate application of 1,3 Dichloropropene 93% [27] prevented this problem, whereas green soil amendment (mustard, grown and embedded before HSB) obtained partial result (Figure 6). Soil disinfection gave rise to a significantly higher tuber yielded, compared to the untreated control, and comparable to the yields of the first year of cultivation, while the green soil amendment resulted in an intermediate tuber yield level (Figure 6C).

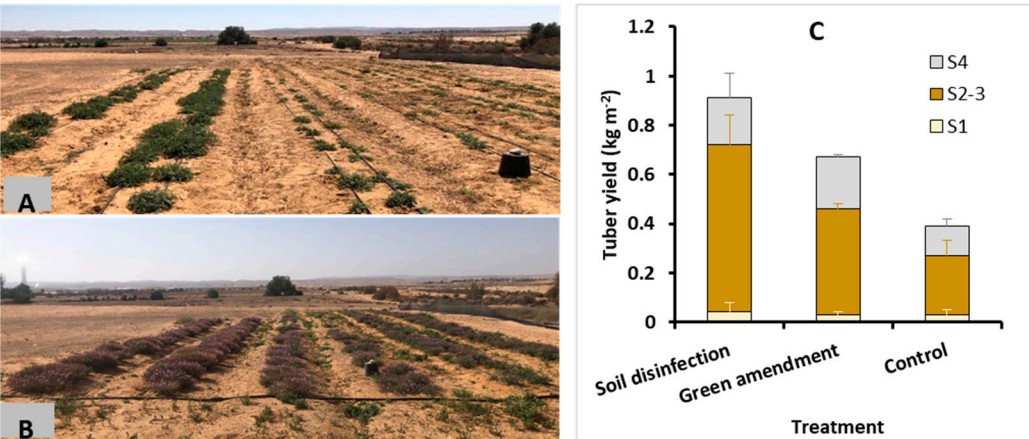

**Figure 6.** The problem of a consecutive HSB cultivation on the same plot. Regular soil disinfection practices supported normal HSB development (**left**), compared to untreated soil beds (**right**). Two (**A**), and 3 (**B**) months after sowing. Effect of soil disinfection and green amendments on yield of HSB tubers grown in consecutive years (**C**).

## 4. Conclusions

Following our demonstration of the pharmaceutical potential of HSB tubers [10,11], the present study describes several essential steps made during the recent decade to start domesticating this wild arid land plant species into an adequate field crop. Exploiting the natural opportunistic features of the species in respect to water availability along with optimizing plant density, the seasonal tuber yield under a generous fertigation regime reached about 10 t ha$^{-1}$. Source-sink manipulations (consistent pruning of the reproductive organs) that almost doubled the tuber yield. An appropriate pest management and effective

soil disinfection treatments that were shown to produce high and consistent HSB tuber yields. Notably, the significant increase in tuber yields due to these introduced agricultural practices was not accompanied by any dilution of the bioactive compounds. Nevertheless, further research and the introduction of new practices are still required to turn HSB plants into an industrial field crop, these include selection and breeding of outstanding clones, fertigation optimization, and the development of various concrete pharmaceutical products.

**Author Contributions:** Conceptualization, S.C., O.G. and H.K.; methodology, A.B., O.G. and M.S.; investigation, O.G., H.K., M.S. and M.M.; formal analysis, M.M., M.S. and O.G.; writing—original draft preparation, A.B., H.K. and O.G.; writing—review & editing, A.B., M.S. and O.G.; funding acquisition, O.G. and H.K. All authors have read and agreed to the published version of the manuscript.

**Funding:** This research was funded by KKL-JNF, the Israeli Ministry of Agriculture and ICA-Israel.

**Conflicts of Interest:** The authors declare no conflict of interest.

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
