# Peer review of "First Steps to Domesticate Hairy Stork’s-Bill (Erodium crassifolium) as a Commercial Pharmaceutical Crop for Arid Regions"

_agronomy, doi:10.3390/agronomy11091715_

Round 1

Reviewer 1 Report

The study entitled “First steps to domesticate hairy stork’s-bill (Erodium crassifolium) as a commercial pharmaceutical crop for arid regions” describes and compares agricultural techniques for the cultivation of a new and potentially pharmaceutical active crop. The presented study of this new crop is quite interesting and describes various steps that have been undertaken to improve potential commercial cultivation of this plant. The authors identified water, fertigation and different pruning techniques to have an effect on tuber yield and anti-inflammatory properties. I find the paper in general well written, however, I do have some concerns that would need to be addressed.

Major concerns:

  1. The authors frequently refer to data that is not shown in the manuscript (personal observation, data not shown, or not mentioning where certain data is obtained from) e.g., line 206, line 214, line 279, line 280-285, line 375, line 387. While I not doubt the general results of the paper, such information needs to be provided either as supplementary material (even preliminary results) or by citing a previous publication. Otherwise, this must clearly be separated from results and be formulated as discussion of the presented results.
  2. The authors mention that “most experiments were conducted using seeds of the ecotype Ramat Negev…”(line 110). I wonder how important the ecotype for the experiments is? The authors compared ecotypes for the anti-inflammatory assays and showed there are differences. It needs to be made clear in what experiments which ecotype was used or it needs to be established that is does not matter which ecotypes is used.
  3. The metabolite composition is a major reason why tubers of hairy stork’s bill are a potentially valuable crop. It would be beneficial for the paper if the authors would describe more concisely which compounds are found and what are the effects observed in the introduction (line 52-74), e.g., describing fractions obtained in a different study are not very helpful for the reader to understand the metabolic composition without knowing how those were separated.
  4. The authors mention that there was variability between different growing seasons analyzed in these papers (which is normal). It would be good if the authors could provide a summary of the observed conditions during the growing periods that were analyzed e.g., in form of a supplementary table.
  5. Figure 2: How many plants were sampled to obtain error bars?
  6. Figure 5: What ecotype was used for plot B and C? Significant differences were tested and replicates were used. Why do the bars not have error bars if replicates were measured?

Minor concerns:

Line 13: would be good to mention why pharmaceutical crop.

Line 66-70: This sounds very speculative without having tested HSB tubers more.

Line 80-84: Can you be more concise here? What techniques exactly accelerate this process? Otherwise, this paragraph is a too vague and not needed.

Line 292-295: I did not understand this sentence. Could you explain?

Line 333: Could you explain here why you chose this assay type and what it is exactly measuring?

Line 335: What is “remarkable” activity? Could you provide some comparisons?

Line 354-358: What could be the explanation for the differences observed between pruned and unpruned HSBs?

Line 400: To what are the authors referring with “bioactivity traits” here?

Author Response

We thank the reviewers for fruitful comments, and believe that the revised manuscript improved significantly and is more concise.

Attached is the letter with all your comments that we addressed in red.

Reviewer 2 Report

This interesting paper describes and discuses different agronomic aspects and agricultural practices with the objective to improving the tuber yield in the novel “crop” Erodium crassifolium in the process of domestication. The agronomic parameters demonstrated a huge work and effort and a very well experimental planning. In the other hand, the production of bioactive compounds in the tubers and their effect on inflammatory response in HCT-L116 cells was analysed in several ecotypes and in response of different nutritional conditions or source:sink manipulations with remarkable  result. In general, the manuscript is clearly written and well-structured,  but some concerns that would need to be addressed.

INTRODUCTION

In the introduction I miss a paragraph about the genetic variability in E. crassifolium (numbers of ecotypes described, differences…) and previous data, if are available, about agricultural practices and growth conditions as crop.

L32.- "Common to" must to be changed by "common in" if you are not sure that HSB is always presented in this ecosystem.

L38.- in is repeat, remove one.

L39.- Paragraph about vegetative growth phase seems to be confused and a reference must be included. I understand that the second phase of vegetative growth (leave and stems extension growth) happens during rainy period.

L41-51.- Please, review if the reference used mainly in this paragraph (8) is correct. Looking for more data about E. crassifolium I have reviewed the chapter from A. Dannin (1988) and any botanical information about the interest sp were found.

L80.- “can profoundly be shortened” could be changed by “can be significantly shortened”

L82.- describing instead of suiting.

MATERIALS AND METHODS

I understand that you sow several seeds per hole to ensure an adequate crops establishment. If more than one seed germinated, Did you leave only one per drip diffuser? .  As you mention in results and discussion L228, nursery-seedlings is the more common system for crops establishment in arid and semiarid environments. You must transplant seedling to the open field and avoid quite heterogeneity in the parameters.

L118.- Please include the dimension of soil beds. I also suggest that include and schematic image about the plots distributions, block design and density. It may make easier to understand.

L119.- Please define seed-nest.

L157.- source:sink instead of source:_sink (homogenize).

L170.- Tuber yield was estimated in Mg/ha. It is a strange unit in agronomy. The production per ha is usually indicated in kg or t.

RESULTS AND DISCUSSION.

In general, the figures with quantitative traits must indicate the sample size (N) and in the figures 3 and 4 a statistical analysis must be done to find significant differences between the different factors analysed, maybe Tukey or Duncan as you have done in figure 5.

L239.- Include a comma instead the middle dash.

L239.- “to the termination” could be change by “the reduction”

L245.- Figure 1 legend. Please normalise the subplot position, maybe at first of the sentence is the best option.

L275.- In this point is discussed that S2-S3 tuber fraction was increase with the nitrate enrichment fertilization but after a deep analysis of the result in figure 3B, I can not detect this difference. By contrast, a clear and interesting increase of S2-S3 tuber fraction was observed when the fertigation was added, almost the increase of yield that you well comment. This point may be discussed.  

L316.- Figure 4, The letter in the subplot (C) is difficult to detect. The data about sink-source experiments shows high variability thus big error bars, if you have a high N you may remove the outlier data. The results suggest that the plants used in this assay were treated with fertigation. Is it true?

L325.- I think that a duplication of S2+S3 tuber fraction is more than a slightly increase of the production, it´s a remarkable increase of the tuber yield in the production scale of this sp.

L331.- In this paragraph or in the introduction you may explain the molecular bases or the mechanism under the inflammatory reaction using HCT-116 cells, TNFalfa and IL-8 quantification. It would help and make easier the reading of the paper for plant biology and agriculture specialist.

Figure 5.- What ecotype of E. crassifolium was used in the experiments show in the subplots 5B and 5C?. Also I miss the error bars in this figure.

L341.- The legend of the figure 5 may be improved. Explain what is represented in each subplot…etc

L346.- remove the gap in ammonium_:nitrate.

Author Response

(The authors gave the same response as above.)
